# A micro-PRNT for the detection of Ross River virus antibodies in mosquito blood meals: A useful tool for inferring transmission pathways

**Narayan Gyawali**⊙*, **Amanda K. Murphy, Leon E. Hugo, Gregor J. Devine**

Mosquito Control Laboratory, QIMR Berghofer Medical Research Institute, Brisbane, Queensland, Australia

\* narayan.gyawali@qimrberghofer.edu.au

## Abstract

### Introduction

Many arboviruses of public health significance are maintained in zoonotic cycles with complex transmission pathways. The presence of serum antibody against arboviruses in vertebrates provides evidence of their historical exposure but reveals nothing about the vector-reservoir relationship. Moreover, collecting blood or tissue samples from vertebrate hosts is ethically and logistically challenging. We developed a novel approach for screening the immune status of vertebrates against Ross River virus that allows us to implicate the vectors that form the transmission pathways for this commonly notified Australian arboviral disease.

### Methods

A micro-plaque reduction neutralisation test (micro-PRNT) was developed and validated on koala (*Phascolarctos cinereus*) sera against a standard PRNT. The ability of the micro-PRNT to detect RRV antibodies in mosquito blood meals was then tested using two mosquito models. Laboratory-reared *Aedes aegypti* were fed, via a membrane, on sheep blood supplemented with RRV seropositive and seronegative human sera. *Aedes notoscriptus* were fed on RRV seropositive and seronegative human volunteers. Blood-fed mosquitoes were harvested at various time points after feeding and their blood meals analysed for the presence of RRV neutralising antibodies using the micro-PRNT.

### Results

There was significant agreement of the plaque neutralisation resulting from the micro-PRNT and standard PRNT techniques ($R^2$ = 0.65; P<0.0001) when applied to RRV antibody detection in koala sera. Sensitivity and specificity of the micro-PRNT assay were 88.2% and 96%, respectively, in comparison with the standard PRNT. Blood meals from mosquitoes fed on sheep blood supplemented with RRV antibodies, and on blood from RRV seropositive humans neutralised the virus by ≥50% until 48 hr post feeding. The vertebrate origin of the

**Data Availability Statement:** All relevant data are within the manuscript.

**Funding:** This work was supported by an Australian Post Graduate Award to A Murphy, by

funds from the Mosquito Control Laboratory, QIMR Berghofer and by the Mosquito and Arbovirus Research Committee (MARC) which is an independent Australian organization funded by local government, government agencies, industry and scientific institutions. The funders had no role in study design, data collection and analysis, decision to publish, or preparation of the manuscript.

**Competing interests:** The authors have declared that no competing interests exist.

blood meal was also ascertained for the same samples, in parallel, using established molecular techniques.

## Conclusions

The small volumes of blood present in mosquito abdomens can be used to identify RRV antibodies and therefore host exposure to arbovirus infection. In tandem with the accurate identification of the mosquito, and diagnostics for the host origin of the blood meal, this technique has tremendous potential for exploring RRV transmission pathways. It can be adapted for similar studies on other mosquito borne zoonoses.

## Introduction

Arthropod borne viruses (arboviruses) present a significant risk to public health globally. In recent decades, rapid urbanization and population growth have assisted the expansion of several viruses from having localised, rural, transmission cycles to being worldwide and urban problems [1]. Epidemiological cycles of many arboviruses, such as Ross River (RRV) and West Nile (WNV) incorporate complex transmission networks involving multiple vertebrate hosts and many vectors. Humans are not necessarily key components of these transmission networks, but increasing human travel, trade and deforestation bring humans into contact with sylvatic/enzootic cycles. This can stimulate arbovirus emergence, re-emergence and spillover into human populations [2–4].

A comprehensive knowledge of the transmission pathways of arboviruses is needed to effectively manage and respond to their emergence. Surveillance systems are needed to identify which mosquito species are responsible for transmission and which animals are acting as amplifying or reservoir hosts. However, the identification of amplifying hosts and transmission pathways remains extremely challenging.

More than 75 arboviruses have been identified in Australia and a small number are associated with human infection [5]. Of these, RRV [6], Barmah Forest virus [7], WNV strain Kunjin [8], and the potentially fatal Murray Valley encephalitis virus [9] are of the greatest public health concern. RRV is the most commonly notified arboviral disease but multiple vectors and many potential vertebrate hosts make this a complex zoonosis. There is little empirical evidence regarding its key transmission cycles or the factors that encourage their spillover to the human population [10, 11].

One means of identifying likely vertebrate disease reservoirs is to demonstrate their historical exposure to disease by searching for virus-specific antibodies in animal sera or tissues. Development of antibody is the major immune response to infection with parasites and pathogens including arboviruses [12, 13]. While such serological evidence of infection does not prove that an animal is an amplifying host or key reservoir, it does allow the generation of hypotheses about probable pathways and is especially useful when combined with information on mosquito species and their host preference. Serological surveys of blood meals are likely to be more fruitful than the direct identification of viruses because vertebrates are only viraemic for a few days, only a small proportion of mosquitoes are virus positive and there is a diminishingly small probability that a captured mosquito will be carrying a virus positive mosquito blood meal. A tremendous sampling effort is therefore required to incriminate reservoir and vector pathways by virus isolation alone.

The potential for screening mosquito blood meals for antibodies to dengue, Japanese encephalitis [14], and WNV [15] has been investigated previously but existing studies required the use of host-specific conjugated antibodies. This is of little utility for the investigation of complex zoonoses like RRV where the hosts are myriad or unknown. The "gold standard" of serological tests is the Plaque Reduction Neutralisation Test (PRNT) [16]. It does not need prior knowledge of host origin but typically requires large quantities of sera or tissue; substantially larger than a typical mosquito blood meal (estimated to be 3 μl [17, 18]).

We developed a micro-PRNT [19, 20] to suit small sample volumes. In this alternative approach, we exploit the fact that vertebrate antibodies persist within mosquito blood meals for some time after the mosquito has fed on a seropositive host. We demonstrate that a micro-PRNT technique can identify vertebrate RRV antibodies in small volumes of sera and mosquito blood meals. This has utility as part of an integrated xenodiagnostic approach that exploits the capture of single blood-fed mosquitoes to infer mosquito species, host preference and host exposure to disease. This will help prioritise potential transmission pathways for further study.

## Materials and methods

### Cells and virus

Vero cells (WHO vaccine strain) and the RRV strain T-48 [21] were obtained from the WHO Collaborating Centre for Arbovirus Reference and Research at the Queensland University of Technology (QUT). At the QIMR Berghofer Medical Research Institute (QIMRB), virus was propagated in Vero cells maintained in 5% $CO_2$ at 37°C in RPMI-1640 growth media (Sigma-Aldrich, Missouri, USA), supplemented with sodium bicarbonate (2 g/L), 10% (v/v) heat-inactivated foetal calf serum (Invitrogen, USA) and 1% (v/v) PSG [(Penicillin (10,000 units)/Streptomycin (10 mg/mL)/ L-glutamine (200 mM)); Sigma-Aldrich, USA]. Virus stocks were frozen at -80°C.

### Koala sera

Forty-two koala sera, obtained from Endeavour Vets, Queensland, Australia (http://www.endeavourvet.com.au) were used to validate the micro-PRNT. These samples were collected between 2015 and 2017 and stored at -80°C.

### Mosquitoes

We used two model insects to validate the micro-PRNT. An *Ae. aegypti* colony that originated from Cairns, Australia, in 2015 and was reared as previously described [22]. Adult mosquitoes were provided with 10% sugar solution *ad libitum* and an opportunity to feed on defibrinated sheep blood once per week. An *Ae. notoscriptus* colony was established from eggs collected in Brisbane, Australia during 2015 and maintained as above.

### Development of the microPRNT

All koala serum samples were tested for neutralising RRV antibodies using a conventional PRNT approach. Equal volumes of sera (200 μl), diluted 1:20 in serum-free RPMI-1640, were mixed with an equal volume of 50 plaque forming units of RRV (1:800 stock RRV in RPMI-1640) per well of a 12-well tissue culture plate (Nunclon, Thermo Scientific, Australia). The virus-sera mixtures were incubated at 37°C for 45 min and added to infect Vero cell monolayers and incubated for a further two hours to enable non-neutralised virus to adsorb to cells. Following incubation, the virus-sera mixture was removed and 2 mL of 0.75% w/v

carboxymethyl cellulose (CMC, Sigma-Aldrich) in RPMI 1640 was added. Plates were incubated at 37°C in 5% v/v $CO_2$ for an additional 40 hr. The CMC/RPMI medium was then removed, and the cell monolayers were fixed and stained with 0.05% w/v crystal violet (Sigma-Aldrich) in formaldehyde (1% v/v) and methanol (1% v/v). The cell monolayers were then rinsed in tap water, and the plates inverted on a paper towel until dry. Plaques (clear zones in a purple cell monolayer) were counted. Reductions of total virus plaque numbers per well of ≥50% were considered to denote seropositive status [13].

All koala samples were also tested by a micro-PRNT technique using just three μl of koala sera; the approximate volume of a mosquito blood meal [17]. Sera were diluted 1:20 in serum-free RPMI-1640. Equal volumes of diluted sera (50 μl) were then mixed with equal volumes of 30 pfu RRV per well (1:160 stock RRV in RPMI-1640) and added to duplicate wells (50 ul per well) of 96–well tissue culture plates (Nunclon, Thermo Scientific, Australia) containing a Vero cell monolayer. A virus density of 30 pfu per well in 96-well plates allowed sufficient visual discrimination of plaques [23]. A volume of 200μl CMC/RPMI was added to each well following infection of the cell monolayer. Plates were scanned at 600 dpi resolution (HP Scanjet, Palo Alto, USA) and images were magnified before counting plaques manually.

The agreement between the percent plaque neutralisation from both the conventional and micro-PRNT was determined by paired sample t-test (n = 42, each serum sample tested once with each PRNT protocol).

## Preparation of blood fed mosquitoes

Blood cells from defibrinated sheep blood (Serum Australis, Manilla, NSW, Australia) were pelleted, washed three times in PBS and then reconstituted in either RRV seropositive or RRV seronegative human sera at a physiological proportion of 1: 0.82 blood cell: plasma. The blood was previously confirmed to be seronegative for RRV by conventional PRNT. Female *Ae. aegypti* mosquitoes (aged 3–5 days) were starved for 5 hr to increase their avidity and then offered seropositive or seronegative blood for 30 min via a membrane feeding apparatus [24]. Fully engorged and unfed mosquitoes from each treatment group were maintained separately at 27±1°C and 80% relative humidity and provided with 10% sucrose solution *ad libitum*. At 6, 12, 24, 36, 48, 60 and 72 hr post exposure to the infected and uninfected blood meals, fed and unfed mosquitoes (the latter used as controls) were harvested and stored at -80 °C.

In a separate experiment, to validate the micro-PRNT on blood meals from mosquitoes that had fed directly on vertebrate hosts, *Ae. notoscriptus* were fed on the exposed arms of RRV seropositive and seronegative human volunteers (previously confirmed by conventional PRNT) for a period of 15 min. Fully engorged mosquitoes were selected for analysis, maintained and harvested as above. Remnant blood could be observed in mosquito abdomens until 60 hours (Fig 1).

Informed, written consent was given for collection of volunteer blood samples and the direct feeding of mosquitoes on sero-negative and sero-positive humans (QIMR Berghofer Human Research Ethics approval P2273).

## Validation of the micro-PRNT using blood-fed mosquitoes

The blood meal volume obtained from a single field-collected, blood-fed mosquito is sufficient for the micro-PRNT assay, but in order to ascertain the assay's robustness against a range of host antibody titres and post-feeding times, and facilitate the dilutions that these experiments required, we used larger volumes of mosquito-derived blood in our validations. These were obtained by pooling three blood meals from engorged mosquitoes that had fed on the same antibody-positive source. One pool was used for each post-feeding time point tested.

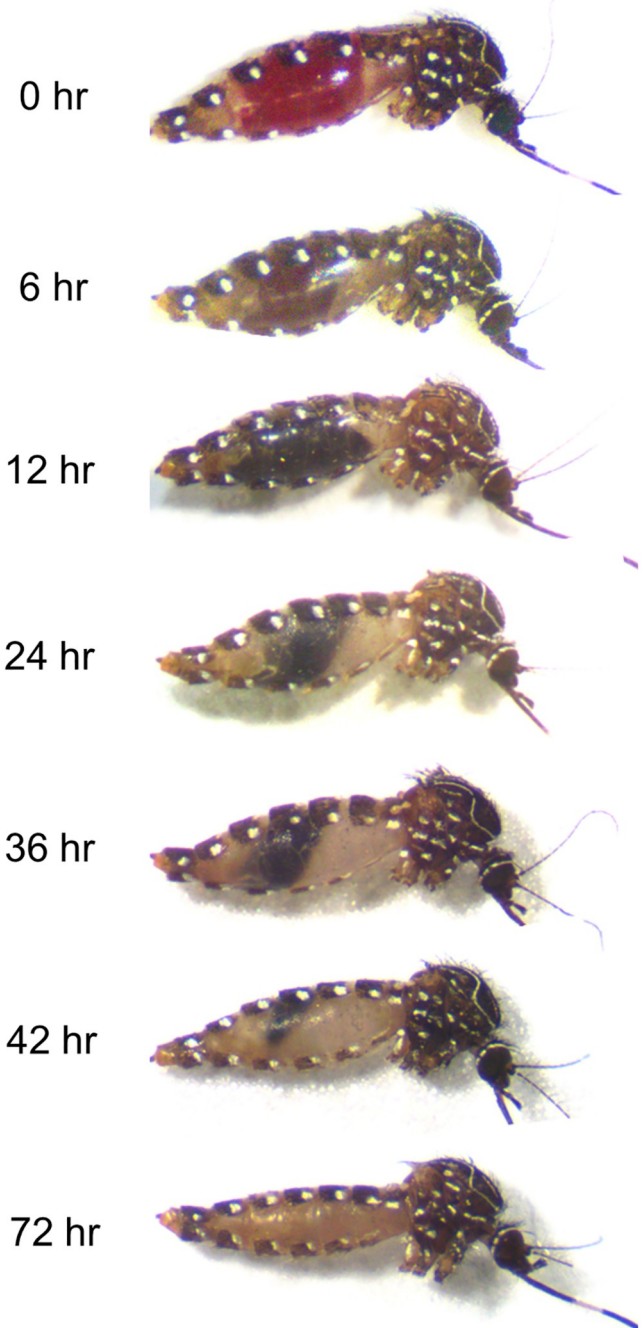

**Fig 1. *Aedes notoscriptus* at different stages of blood meal digestion.** Immediately after feeding (0 hr), mosquitoes were fully engorged. After 72 hr, the blood meal could no longer be seen.

Abdominal contents were diluted by 1:20, 1:40, 1:80 and 1:160 using serum-free RPMI 1640 supplemented with 1% PSG and 0.4% amphotericin B (Sigma-Aldrich, USA). The final reported blood volume was calculated based on dilution of a 3 μl blood meal [17] [18]. A range of controls were also processed to assess the impacts of sero-negative mosquito homogenates on the inhibition of plaque forming units. This entailed a comparison of 1) RRV alone 2) RRV

plus unfed mosquito abdomens, and 3) RRV plus the abdomens of mosquitoes that contained sheep blood supplemented with seronegative human sera. In every case, the RRV pfu was kept constant.

To validate the ability of the micro-PRNT to detect vertebrate anti-RRV antibodies from single, blood fed mosquitoes, we tested *Aedes notoscriptus* that had been fed on human seropositive or seronegative volunteers. Blood-fed mosquitoes were harvested at various times post-feeding and processed as above (abdominal contents expelled into serum-free RPMI 1640 and adjusted to obtain a 20-fold dilution). Unfed mosquitoes were included as a control. Each 96-well plate was scanned and plaques were counted manually as described above.

## Identification of host DNA

Sixteen blood meal samples harvested after feeding on RRV seropositive sheep blood or human blood (n = 8 for each) were used to demonstrate that host identification could be performed in parallel with the micro-PRNT on the same blood samples. PCR amplification of *Cytochrome b* was performed as previously described [25].

## Results

### Comparison of micro-PRNT and PRNT

RRV plaques on cell layers stained 40 hr post-incubation were clearly distinguishable and easily counted when plates were scanned and images enlarged. There was a significant correlation ($R^2$ = 0.65; P<0.0001) between percent neutralisation of RRV pfu noted in koala samples characterized by micro-PRNT or PRNT techniques (Fig 2). In comparison to the standard PRNT, the sensitivity and specificity of the micro-PRNT was 88.2% and 96% respectively. Those differences in percent neutralisation determined between methods were not significant ($p > 0.05$; paired samples t-test).

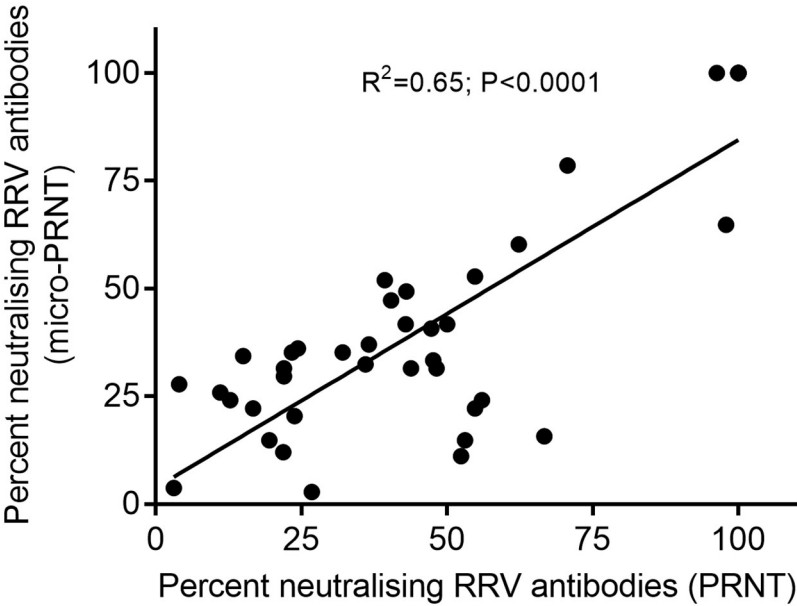

**Fig 2. Plaque neutralisation demonstrated by standard PRNT or micro-PRNT using koala serum samples.**
Correlation of percent reduction in plaque forming units as measured by standard PRNT (x-axis) and micro-PRNT (y-axis) techniques.

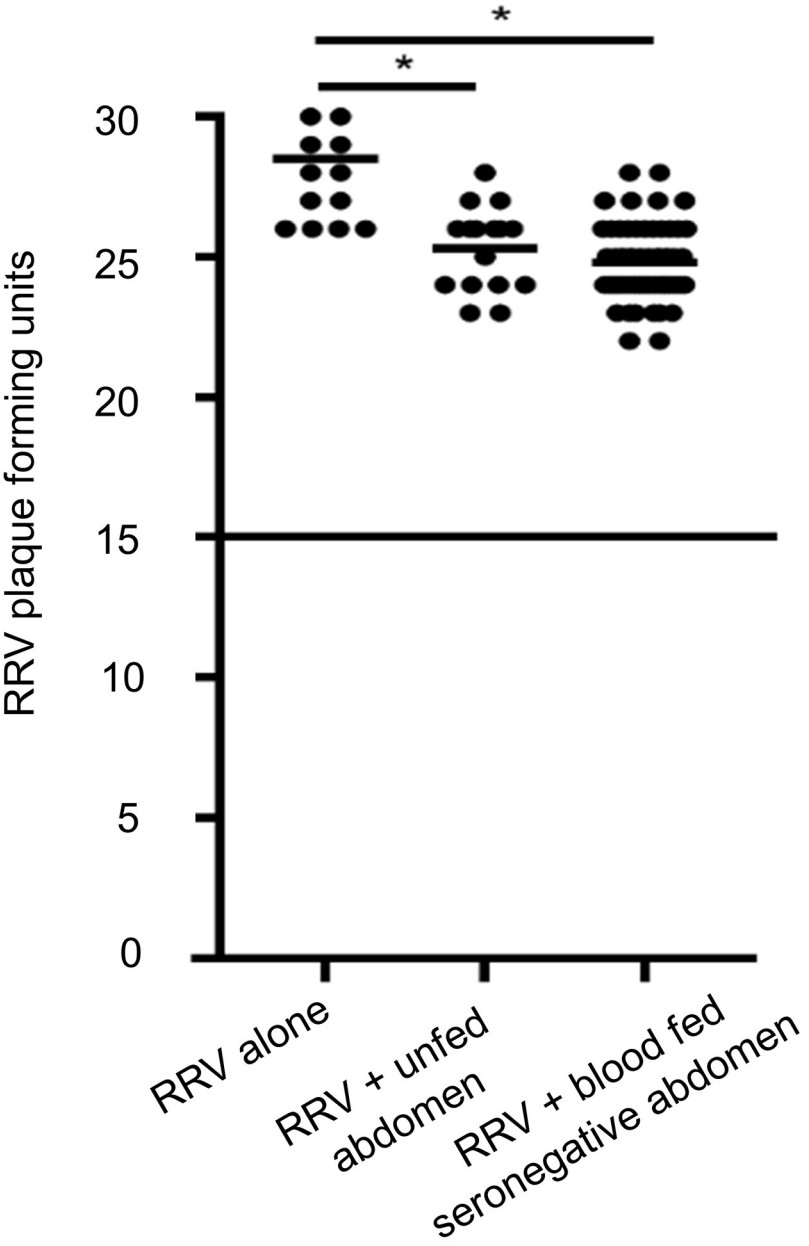

**Fig 3. The effect of mosquito homogenates on plaque formation.** Homogenates of mosquito abdomens reduce the number of plaque forming units by approximately 10%, when compared with Vero cells inoculated with RRV alone (*p<0.05, calculated by one-way ANOVA test).

### Impacts of mosquito homogenates on the number of plaque forming units

Vero cells were inoculated with RRV alone, RRV mixed with homogenates of unfed mosquito abdomens and homogenates of abdomens containing RRV seronegative sheep blood. There was a small (10%) but significant decrease in RRV pfu when Vero cells were inoculated with RRV mixed with the latter two samples (Fig 3).

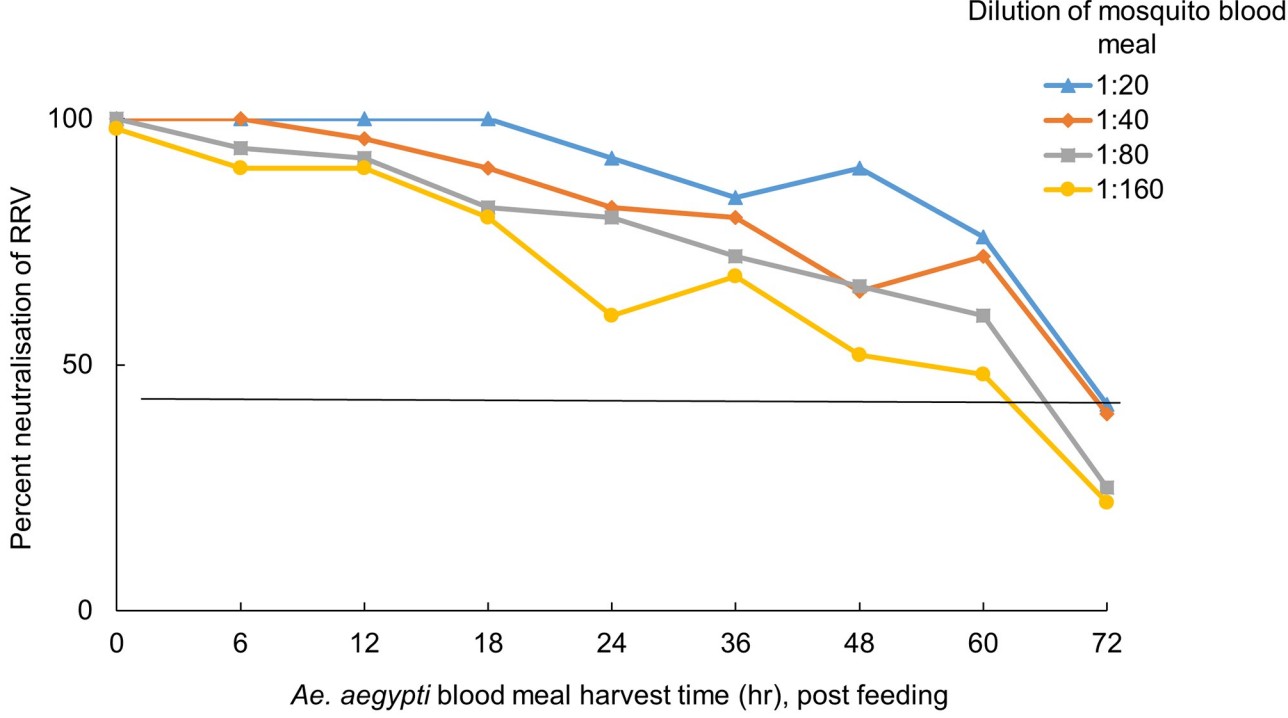

**Fig 4. Percent neutralisation of RRV by vertebrate antibodies in mosquito blood meals harvested at different time points post blood-feeding.** Blood meals obtained from RRV positive sera and harvested at different times post-feeding were diluted at four different concentrations and tested in duplicate. Antibodies continued to neutralise ≥ 50% of plaques at all dilutions, until 60 hr post feeding.

### Validation of the assay using mosquito blood meals

Mosquito blood meals obtained from mosquitoes membrane-fed on sheep blood mixed with anti-RRV human antibodies neutralised RRV by ≥50% until 60 hr post blood feeding (Fig 4). In contrast, there was no neutralisation of blood meals obtained from mosquitoes fed with sheep blood supplemented with RRV seronegative human serum. These assays demonstrated that the micro-PRNT is robust across a range of dilutions that are likely to represent varying antibody levels in the host.

Guided by the results detailed in Fig 4, we used the 1:20 dilution for all subsequent micro-PRNTs on single blood meals from live hosts. Single *Ae. notoscriptus* blood meals obtained by feeding mosquitoes on a seropositive human volunteer were harvested at different time points. These blood meal preparations neutralised RRV pfu by ≥50% until 48 hr post feeding. (Fig 5A). The limited neutralising effects of sero-negative blood meals (Fig 5B) and un-fed mosquito abdomens are included for comparison (Fig 5C).

### Host identification of blood meal samples

Sequencing of *cytochrome b* amplicon-DNA from all 16 blood meal samples correctly identified (>95% nucleotide identity to *cytochrome b* sequences) the origin of the blood meal source (i.e. sheep from those experiments that had used membrane feeds, and human where mosquitoes had fed on volunteers).

### Discussion

Mosquito blood meals are a potentially useful resource for assessing antibody seroprevalence in vertebrates and inferring the RRV transmission pathways between vectors, disease

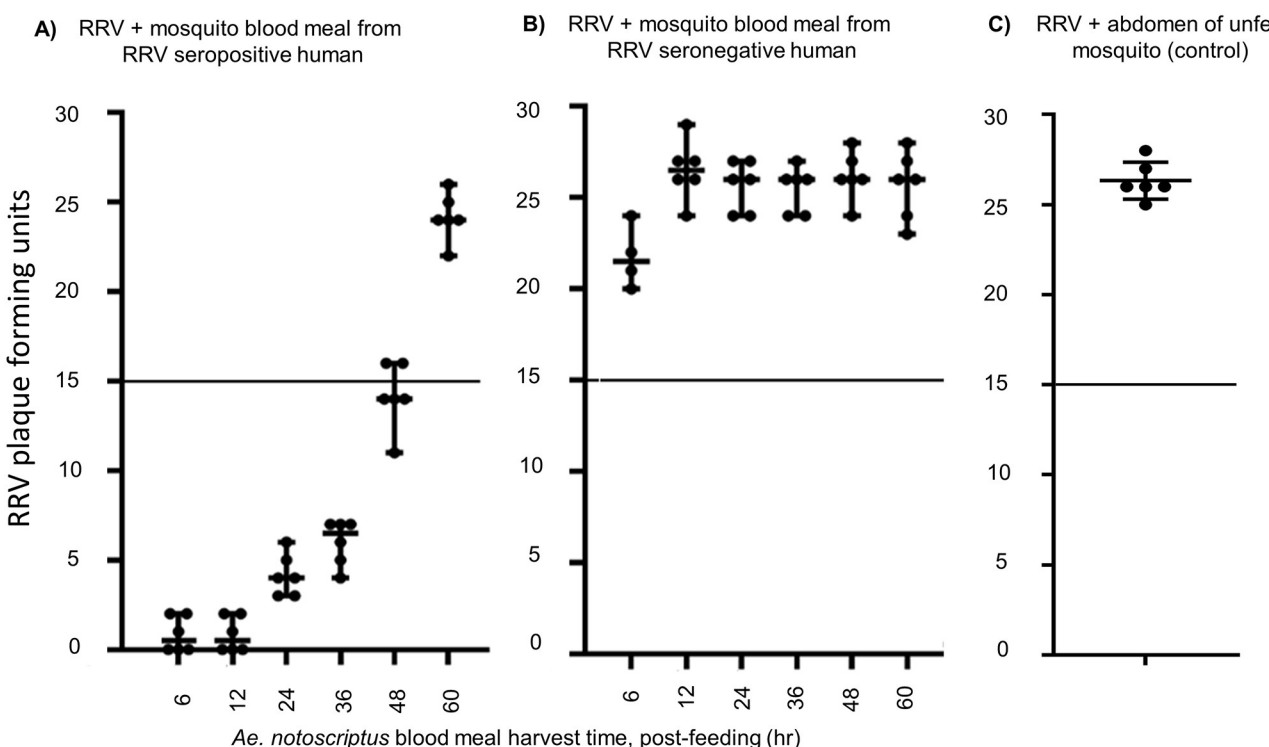

**Fig 5. Impact of post-feeding times on neutralisation of RRV.** Blood meals from an RRV seropositive human volunteer (A), blood meals from an RRV seronegative human volunteer (B) and unfed mosquito abdomens as a control (C). Antibody positive blood meals continued to neutralise ≥ 50% of plaques until 48 hours post-feeding.

reservoirs and humans. This study demonstrates that a micro-PRNT using 96-well plates has considerable utility for characterizing the antibody content of small blood volumes and is equivalent in sensitivity and specificity to the "gold standard" conventional PRNT method. In conjunction with the species identification of the blood-fed mosquito, and the use of existing molecular tools to identify the host origin of the blood meal, this new diagnostic has the capacity to increase our understanding of the key pathways for the transmission of complex zoonotic arboviruses.

The accuracy of the micro-PRNT for testing RRV antibodies in this study compared favourably with a conventional PRNT method when applied to our koala samples. Similarly, confluent results were reported for a micro-PRNT tested against yellow fever virus in artificially spiked serum samples [19]. Although alternatives to the PRNT such as a VecTest-inhibition assay and a biotin microsphere immunoassay have been used to identify pathogen-specific antibodies in mosquito blood meals [15]; both require host-specific antibodies and the latter demands considerable laboratory resources. Our micro-PRNT demonstrates sufficient sensitivity, specificity and utility for the determination of RRV antibodies in blood-fed mosquitoes that have fed on any vertebrate.

Our micro-PRNT could detect ≥50% RRV neutralisation by mosquito blood meals up to 36–48 hr post blood-feeding. Although this is the first study to have identified RRV antibodies in mosquito blood meals, more general studies have shown that antibodies can survive in those environments. Hatfield (1988) identified Bovine Serum Albumin (BSA) specific antibodies using antibody-captured ELISA from the hemolymph of *Ae. aegypti* up to 48 hr after feeding [26]. Irby and Apperson (1989) used an immunoblot technique to demonstrate that serum

proteins from rodents and humans persisted in *Ae. aegypti* blood meals for 36 to 48 hr post-feeding [27]. Anti-BSA antibodies were detected 9 days after blood feeding in *Anopheles stephensi* [28] and human specific IgM and IgG were present in the blood meals of *Ae. albopictus* for 7 days [29]. These extended periods are surprising given that one would expect blood meals and their proteins to have been fully digested by then but the persistence of antibodies in mosquito blood meals will differ with species, ambient temperature, body size, initial concentration of antibodies, blood meal volume and the length of the gonotrophic cycle. In our study, the period of detectability (48–60 hr) corresponded to the period that blood was externally visible in the abdomen (Fig 1).

The small volumes of homogenate left from a single mosquito after execution of the micro-PRNT allows for a parallel PCR amplification for identification of the blood meal source (the host). The literature commonly reports that the origin of host blood meals can be identified from as a little as 0.02 μl of blood [30]. In our proof of principle, 10 μl aliquots of 1:20 diluted homogenate recovered from the micro-PRNT 60 hr post blood-feeding were successfully amplified and sequenced to identify our experimental donors: sheep and humans.

Given the challenges involved in obtaining, trapping and screening wild animals for serum sampling and sero-prevalence studies [13], the collection of blood fed mosquitoes is a useful means of sampling, with mosquitoes acting as an indirect sampling tool or "flying syringe" for sampling inaccessible or ethically challenging hosts. In terms of pathway incrimination, a single mosquito will yield information on vectors, host preference and the disease exposure of that host [11]. That information, especially when combined with risk modelling [31] can be used to identify those transmission pathways of greatest importance within the host community. Various studies have observed the potential for insect blood meals to detect pathogen specific antibodies [26, 32–34], but none have developed high throughput methods suitable for application to transmission ecologies involving unknown reservoirs.

Human health can be affected by infectious diseases of wildlife living close to human habitation. The risks are increasingly common in Australia and elsewhere because of increasing encroachment of the human population on diverse mosquito habitats and the adaptation of pathogen reservoir species to urbanized environments [35, 36]. Dengue, Hantavirus, Lyme disease, Zika, avian influenza, and rabies are examples of globally endemic zoonoses that have emerged from human encroachment into rural or sylvatic habitats [37, 38]. Similarly, RRV is a major public health risk in Australia, maintained in a diverse range of hosts and vectors and undergoing an expansion in range to the Pacific Islands [39].

Infectious diseases are also a concern for wildlife conservation, particularly those already threatened by habitat loss and exploitation. Surveillance of wild animals for infection or disease commonly involves trapping or killing animals for direct sampling of blood and tissues. This can be difficult, expensive, dangerous and sometime unethical. The technique demonstrated here is not only applicable to RRV reservoir identification but also to other arboviruses and infectious agents which have complex transmission cycles and a range of vertebrate hosts.

There was a ≈10% inhibition of virus pfu by mosquito tissue homogenates (Fig 3). However, this inhibition was minimal compared to the threshold used to determine positive neutralisation (>50% reduction). One possible explanation of this observation is that some component of abdominal tissue may have an inhibitory effect on virus replication.

As for all serum or tissue collection techniques, reliance on blood-fed mosquitoes as a sampling tool will be subject to sampling bias. Mosquitoes may be differentially attracted to diseased hosts [40] or to species that are uncharacteristically abundant at any single point in time. Different mosquito trapping techniques have differential vector specific targets, so using one particular trap type might miss key vector species. In terms of serology, there may be considerable cross reaction between some virus antibodies [12]. In this case RRV is likely to cross-react

with the closely related alphavirus, Barmah Forest virus, whose epidemiology in Queensland remains significant [2]. Finally, those reservoirs inferred by blood meal analysis may not be the key amplifying host, and the mosquitoes incriminated may not be the key vector. For example, dengue antibodies are commonly found in *Culex* spp. mosquitoes during epidemics, but those mosquitoes do not transmit the disease. Nonetheless their blood meals may still identify the host and its sero-prevalence rate [14]. For all of these reasons, the various components of the pathways implicated by the micro-PRNT technique and attendant host identifications must be interpreted and prioritised in the light of all the available knowledge on the ecology of the disease.

## Conclusions

The value of the micro-PRNT lies in its ability to detect anti-virus antibodies from mosquito blood meals obtained from any vertebrate host. When coupled with molecular identification of the host by DNA amplification and sequencing, valuable information on vector-host relationships, wildlife sero-prevalence rates and zoonotic transmission cycles can be inferred [14]. This novel xenodiagnostic offers an alternative "flying syringe" approach for serum sampling and for monitoring sero-prevalence in animals. The use of this assay to characterise the blood meals of mosquitoes collected from the field in Brisbane is now underway.

## Acknowledgments

We thank Prof. John Aaskov and Dr. Francesca Frentiu from QUT for their advice and for the supply of Vero cells and the RRV isolate.

## Author Contributions

**Conceptualization:** Narayan Gyawali, Amanda K. Murphy, Leon E. Hugo, Gregor J. Devine.

**Funding acquisition:** Gregor J. Devine.

**Methodology:** Narayan Gyawali, Amanda K. Murphy, Leon E. Hugo.

**Supervision:** Gregor J. Devine.

**Validation:** Narayan Gyawali.

**Writing – original draft:** Narayan Gyawali.

**Writing – review & editing:** Narayan Gyawali, Amanda K. Murphy, Leon E. Hugo, Gregor J. Devine.

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
