## [Decision Letter · Decision Letter 0]

5 May 2020

PONE-D-20-02979

A micro-PRNT for the detection of Ross River virus antibodies in mosquito blood meals: a useful tool for inferring transmission pathways

PLOS ONE

Dear Dr. Gyawali,

Thank you for submitting your manuscript to PLOS ONE. After careful consideration, we feel that it has merit but does not fully meet PLOS ONE’s publication criteria as it currently stands. Therefore, we invite you to submit a revised version of the manuscript that addresses the points raised during the review process.

Firstly, sorry for taking too long for this review. It was very difficult to secure reviewers. Please carefully respond, point by point, to all queries raised by both reviewers.

We would appreciate receiving your revised manuscript by June15th. To enhance the reproducibility of your results, we recommend that if applicable you deposit your laboratory protocols in protocols.io, where a protocol can be assigned its own identifier (DOI) such that it can be cited independently in the future. For instructions see: http://journals.plos.org/plosone/s/submission-guidelines#loc-laboratory-protocols

We look forward to receiving your revised manuscript.

Kind regards,

Luciano Andrade Moreira, PhD

Academic Editor

PLOS ONE

Journal Requirements:

1. Thank you for including your ethics statement; "QIMR Berghofer Medical Research Institute ethics ref: P2273. "

Please amend your current ethics statement to confirm that your named ethics committee Institutional Care and Use Committee (IACUC) specifically approved this study.

For additional information about PLOS ONE ethical requirements for animal research, please refer to http://journals.plos.org/plosone/s/submission-guidelines#loc-animal-research

2. Please provide additional details regarding participant consent for participation in mosquito feeding. In the ethics statement in the Methods and online submission information, please ensure that you have specified (1) whether consent was informed and (2) what type you obtained (for instance, written or verbal). If your study included minors, state whether you obtained consent from parents or guardians. If the need for consent was waived by the ethics committee, please include this information

Reviewers' comments:

Reviewer's Responses to Questions

**Comments to the Author**

1. Is the manuscript technically sound, and do the data support the conclusions?

Reviewer #1: Yes

Reviewer #2: Yes

2. Has the statistical analysis been performed appropriately and rigorously? 

Reviewer #1: I Don't Know

Reviewer #2: Yes

3. Have the authors made all data underlying the findings in their manuscript fully available?

Reviewer #1: No

Reviewer #2: Yes

4. Is the manuscript presented in an intelligible fashion and written in standard English?

Reviewer #1: Yes

Reviewer #2: Yes

5. Review Comments to the Author

Reviewer #1: Interesting technical paper with only minor edits.

line 70 use of word "encourage" needs to be reexamined

line 91 West Nile viruses

lines 233 and 241 missing period.

line 302 sometimes

Overall - I am not sure how useful some of the figures are

Figure 1 - need T=0 or state it was fully engorged - even with this I am not sure it really adds anything

Figure 4 B and C - so small hard to see better to summarize data and leave the plates out?

Reviewer #2: The work proposes the use of micro PRNT from blood present in mosquito abdomens, useful tool for viral ecology projects, standardized in that study for Ross River virus. Very important and well-designed work.

Line 96 – What is "ca"? Appears on line 305 again.

Line 107 – The reagent "L-glutamine (200 mM)" is repeated in the formulation. I believe it must be an antimicrobial, perhaps Gentamicin.

Lines 171 – 172 “The volume of RPMI 1640 was adjusted to obtain a 20-fold dilution of the blood.” What level of volume control is there in this process? Is the blood volume measured? That part was not clear.

Lines 180 – 181 “Aedes notoscriptus that had been fed on human seropositive or seronegative volunteers” Mention the quality control that is done on these mosquitoes, as they were used to feed on human volunteers.

The legends of figures 1, 2 and 3 are incomplete. In all of them the information needs to be complemented.

Discussion

There is a lack of discussion topic of the complexity of using the tool in fieldwork regarding the possible species involved as hosts. According to the protocol described here, the volume would be enough to screen how many species? Does the protocol allow a focused or broad-spectrum host approach? The proof of concept was made with blood obtained from human and sheep, but how would be the use of the protocol in other works, considering that this is the final objective of the publication.

6. PLOS authors have the option to publish the peer review history of their article (what does this mean?). If published, this will include your full peer review and any attached files.

Reviewer #1: No

Reviewer #2: Yes: Pedro Augusto Alves

---

## [Author Response · Author response to Decision Letter 0]

5 Jun 2020

Manuscript ID PONE-D-20-02979 Response to Reviewers

Title: A micro-PRNT for the detection of Ross River virus antibodies in mosquito blood meals: a useful tool for inferring transmission pathways.

Comments to the Editor

We have amended the ethics statements according to your instructions. Written consent was required by volunteers. That information is now included in the last paragraph of the section “Preparation of blood fed mosquitoes”.

We have made a number of changes to the text, additional to the requests may by the reviewers. These are visible in the track changed R1 manuscript. There have been no changes in context, methodology or interpretation but we have altered sections of text for clarity or to correct grammar. The major changes are detailed below:

1. Introduction: We have moved two paragraphs form the end of the introduction to further up the page.

2. Preparation of blood fed mosquitoes: First and last paragraphs revised for methodological clarity and to include requested information on ethics. 

3. Validation of the micro PRNT: We have clarified some of information on blood meal volumes.

4. The text in the discussion has been improved for flow and clarity, but the interpretation of results has not changed. 

Response to Reviewers

Reviewer #1

Comment 1: Interesting technical paper with only minor edits.

Line 70 use of word "encourage" needs to be re-examined

Line 91 West Nile viruses

Lines 233 and 241 missing period.

Line 302 sometimes

Response: We thank the reviewer for their positive assessment of the study and for identifying these minor edits. These have been corrected in the updated manuscript.

Comment 2: Overall - I am not sure how useful some of the figures are. Figure 1 needs T=0 or state it was fully engorged - even with this I am not sure it really adds anything.

Figure 4 B and C - so small hard to see better to summarize data and leave the plates out?

Response: We feel that Fig 1 is a useful visual representation of the amount of blood present at various times post feeding. We have updated Fig 1 to include 0 (fully engorged) and 6h post feeding. 

For Fig 4, we agree with the reviewer. We have removed the photos of the micro-titre plates, leaving only the line graph to demonstrate the relationship between antibody concentration and plaque inhibition over time.

Reviewer #2

Comment 1: The work proposes the use of micro PRNT from blood present in mosquito abdomens, useful tool for viral ecology projects, standardized in that study for Ross River virus. Very important and well-designed work.

Line 96 – What is "ca"? Appears on line 305 again.

Line 107 – The reagent "L-glutamine (200 mM)" is repeated in the formulation. I believe it must be an antimicrobial, perhaps Gentamicin.

Lines 171 – 172 “The volume of RPMI 1640 was adjusted to obtain a 20-fold dilution of the blood.” What level of volume control is there in this process? Is the blood volume measured? That part was not clear.

Lines 180 – 181 “Aedes notoscriptus that had been fed on human seropositive or seronegative volunteers” Mention the quality control that is done on these mosquitoes, as they were used to feed on human volunteers.

The legends of figures 1, 2 and 3 are incomplete. In all of them the information needs to be complemented.

Response: We thank the reviewer for their positive assessment of the study. 

“ca” (Circa) has been replaced with “approximately”. 

We have removed reference to L-glutamine which is an amino acid and a common additive to cell culture media.

We used fully engorged mosquitoes, published estimates of mean blood volumes (e.g Konishi 1989, Jeffery 1956) and applied dilution factors to those published estimates. 

All mosquitoes were from established, uninfected colonies. The sero-status of human volunteers was confirmed by standard PRNT. Each blood-feed lasted until engorgement or termination (30 minutes). Only mosquitoes that were fully engorged were selected for further analysis. Three engorged mosquitoes per time point were chosen for analysis. 

We have attempted to make this process clearer in the methods section under ‘Preparation of blood fed mosquitoes’.

The legends for figures 1, 2 and 3 have been expanded.

Comment 2: There is a lack of discussion topic of the complexity of using the tool in fieldwork regarding the possible species involved as hosts. According to the protocol described here, the volume would be enough to screen how many species? Does the protocol allow a focused or broad-spectrum host approach? The proof of concept was made with blood obtained from human and sheep, but how would be the use of the protocol in other works, considering that this is the final objective of the publication.

Response: We believe that we have already acknowledged the complexity in interpretation in our discussion, having referred to sampling bias, cross-reactivity, inability to identify all potential vertebrate hosts and the fact that the presence of antibody in a mosquito blood meal does not necessarily define that mosquito as a key vector. 

Our proof of principle shows that the same mosquito blood meals can be used to identify the host origin of the blood meal and whether that host had been exposed to a target virus (in this case the arbovirus RRV). The technique provides clues to potential transmission pathways, but is not definitive.

We agree with the reviewer that a broad spectrum of hosts might be represented in mosquito blood meals. Although we used human and sheep blood for demonstration purposes, sequencing data from a single, tiny aliquot of serum can be matched to myriad vertebrate sequences available online.

---

## [Decision Letter · Decision Letter 1]

8 Jul 2020

A micro-PRNT for the detection of Ross River virus antibodies in mosquito blood meals: a useful tool for inferring transmission pathways

PONE-D-20-02979R1

Dear Dr. Gyawali,

We’re pleased to inform you that your manuscript has been judged scientifically suitable for publication and will be formally accepted for publication once it meets all outstanding technical requirements.

Kind regards,

Luciano Andrade Moreira, PhD

Academic Editor

PLOS ONE

Additional Editor Comments (optional):

Reviewers' comments:

Reviewer's Responses to Questions

**Comments to the Author**

1. If the authors have adequately addressed your comments raised in a previous round of review and you feel that this manuscript is now acceptable for publication, you may indicate that here to bypass the “Comments to the Author” section, enter your conflict of interest statement in the “Confidential to Editor” section, and submit your "Accept" recommendation.

Reviewer #1: All comments have been addressed

Reviewer #2: All comments have been addressed

2. Is the manuscript technically sound, and do the data support the conclusions?

Reviewer #1: Yes

Reviewer #2: Yes

3. Has the statistical analysis been performed appropriately and rigorously? 

Reviewer #1: I Don't Know

Reviewer #2: Yes

4. Have the authors made all data underlying the findings in their manuscript fully available?

Reviewer #1: Yes

Reviewer #2: Yes

5. Is the manuscript presented in an intelligible fashion and written in standard English?

Reviewer #1: Yes

Reviewer #2: Yes

6. Review Comments to the Author

Reviewer #1: Thank you for the edits, this reads much better now and the figures presented are more appropriate. This is an interesting technique that can potentially be modified and used for other arboviruses.

There are 2 missing a periods in lines 239 and 246 at the end of the paragraphs.

Reviewer #2: (No Response)

7. PLOS authors have the option to publish the peer review history of their article (what does this mean?). If published, this will include your full peer review and any attached files.

Reviewer #1: No

Reviewer #2: No

---

## [Editor Report · Acceptance letter]

13 Jul 2020

PONE-D-20-02979R1 

A micro-PRNT for the detection of Ross River virus antibodies in mosquito blood meals: a useful tool for inferring transmission pathways 

Dear Dr. Gyawali:

I'm pleased to inform you that your manuscript has been deemed suitable for publication in PLOS ONE. Congratulations! Your manuscript is now with our production department. 

Kind regards, 

on behalf of

Dr. Luciano Andrade Moreira 

Academic Editor

PLOS ONE